# Ab Initio Rovibrational Spectroscopy of the Acetylide Anion

**DOI:** 10.3390/molecules28155700

**Published:** 2023-07-27

**Authors:** Benjamin Schröder

**Affiliations:** Institute of Physical Chemistry, University of Goettingen, Tammannstr. 6, 37077 Göttingen, Germany; bschroe4@gwdg.de

**Keywords:** acetylide anion, potential energy surface, variational calculations, rovibrational spectrum, spectroscopic parameters

## Abstract

In this work the rovibrational spectrum of the acetylide anion HCC${}^{-}$ is investigated using high-level electronic structure methods and variational rovibrational calculations. Using a composite approach the potential energy surface and dipole surface is constructed from explicitly correlated coupled-cluster accounting for corrections due to core-valence correlation, scalar relativistic effects and higher-order excitation effects. Previous approaches for approximating the latter are critically evaluated. Employing the composite potential, accurate spectroscopic parameters determined from variational calculations are presented. In comparison to the few available reference data the present results show excellent agreement with ground state rotational constants within 0.005% of the experimental value. Intensities determined from the variational calculations suggest the bending fundamental transition $\nu_{2}$ around 510 cm${}^{-1}$ to be the best target for detection. The rather weak CD stretching fundamental $\nu_{1}$ in deuterated isotopologues show a second-order resonance with the $\left(0,\;2^{0},\;1\right)$ state and the consequences are discussed in some detail. The spectroscopic parameters and band intensities provided for a number of vibrational bands in isotopologues of the acetylide anion should facilitate future spectroscopic investigations.

## 1. Introduction

Many of the ca. 240 molecules which have been detected in the interstellar medium or circumstellar shells [1,2,3,4,5] are highly reactive species such as radicals, carbenes or molecular ions. These can be challenging to study in the laboratory by spectroscopic techniques due to low obtainable concentrations. Therefore, highly accurate predictions of spectroscopic parameters based on ab initio theory are desirable [6,7,8,9,10,11,12]. While about 10% of the astromolecules are cationic species, the number of detected anions amounts to only a quarter of that [5]. See also the work of Millar et al. [13] for reviews on the role of anions in astrophysics. Among the anionic interstellar molecules the class of linear chains of type C${}_{2n}$H${}^{-}$ is the most extensive with C${}_{6}$H${}^{-}$ being the first molecular anion to ever be detected in the interstellar medium [14,15,16,17,18,19].

Following the initial detection as a series of unidentified lines (termed B1377) toward IRC + 10216 by Kawaguchi et al. [14] it took more than 10 years until McCarthy and coworkers [15] could record the rotational spectrum of C${}_{6}$H${}^{-}$ in the centimeter- and millimeter-wave bands and unambigously assign the B1377 transition. This lead to an increased effort of detecting more anionic species and the series was extended in the following year through the work of Cernicharo and coworkers [18] observing C${}_{4}$H${}^{-}$ towards IRC + 10216 as well as Brünken et al. [16] and Remijan et al. [17] who independently reported the detection of C${}_{8}$H${}^{-}$. Quite recently Remijan and coworkers [19] published results as part of their GOTHAM program (GBT Observations of TMC-1: Hunting Aromatic Molecules) describing the identification of C${}_{10}$H${}^{-}$ towards TMC-1 using the Green Bank Telescope.

Despite considerable effort [20,21], the acetylide anion HCC${}^{-}$—the smallest member of the series—has not yet been detected in space. This has been ascribed to a reduced rate for the dominant radiative electron attachment path in the fromation of these anions [22,23,24]
$$C_{n}H+e^{-}\to C_{n}H^{-}+h\nu \;.$$

Furthermore, spectroscopic information for this system is rather scarce. In 2007 Brünken et al. [25] succeeded in recording the rotational spectrum of HCC${}^{-}$ produced by discharge through an acetylene/argon mixture in a free space millimeter-wave spectrometer. The five lowest rotational transitions of HCC${}^{-}$ were observed as well as rotational transitions of the two singly ${}^{13}$C substituted species. One year later Amano [26] extended the range of observed rotational lines for the main isotopologue up to the 10←9 transition by the use of a submillimeter-wave spectrometer and extended negative glow discharge. Together these measurements provide accurate vibrational ground state rotational parameters $B_{0}$ and $D_{0}$ of 41,639.23501(94) MHz and 96.9039(62) kHz, respectively, where numbers in parentheses indicate one standard deviation to the last significant digits. With regards to the vibrational spectrum of HCC${}^{-}$ the only reliable available experimental information has been obtained by Ervin and Lineberger [27] from photoelectron spectroscopy of HCC${}^{-}$. Through the analysis of hot bands in the HCC(${\tilde{X}}^{2}\Sigma^{+}$) ← HCC${}^{-}$(${\tilde{X}}^{1}\Sigma^{+}$) spectrum they determined the bending fundamental frequency to be $\nu_{2}\;=\;505\pm 10$ cm${}^{-1}$ and the CC-stretching fundamental $\nu_{3}\;=\;1800\pm 20$ cm${}^{-1}$.

Given its importance there have been a number of theoretical studies dedicated to the spectroscopy of HCC${}^{-}$ [28,29,30,31,32,33,34]. A careful investigation was presented by Mladenović et al. [32] who developed an analytic three-dimensional representation of the potential energy surface (PES) based on coupled-cluster calculations with singles, doubles, and perturbative triples CCSD(T) [35]. Variational calculations using that PES yielded band origins of the fundamental vibrational transitions of 511 and 1805 cm${}^{-1}$ for $\nu_{2}$ and $\nu_{3}$, respectively, as well as a value of 3211 cm${}^{-1}$ for the CH-stretching fundamental $\nu_{1}$. These results also ruled out claims by Gruebele et al. [36] who reported the observation of the $\nu_{3}$ band around 1758.621(3) cm${}^{-1}$. The experimentally determined band center clearly is at odds with both the theoretical predictions as well as the results of Ervin and Lineberger [27]. By comparison with results obtained for hydrogen cyanide (HCN) at the same level of theory and experiment [37] Mladenović et al. derived recommended equilibrium bond lengths of $r_{e}\left(\mathrm{CH}\right)\;=\;1.0689\left(3\right)$ Å and $R_{e}\left(\mathrm{CC}\right)\;=\;1.2464\left(2\right)$ Å.

In 2009 Huang and Lee presented results for HCC${}^{-}$ based quartic force fields (QFF) within a composite procedure that combines complete basis set (CBS) limit CCSD(T) results with corrections for missing correlation contributions. Besides core-valence correlation and scalar relativistic effects, emphasis was given to the inclusion of higher-order correlation contributions beyond CCSD(T) which were obtained by forming the difference between averaged coupled-pair functional (ACPF) [38] calculations based on a full-valence complete active space self-consistent field (CASSCF) [39] reference and canonical CCSD(T). The final equilibrium bond lengths were $r_{e}=1.24702$ Å and $R_{e}=1.06967$ Å. Based on vibrational configuration interaction calculations [40,41,42] Huang and Lee obtained $\nu_{1}\;=\;3204$ cm${}^{-1}$, $\nu_{2}\;=\;502$ cm${}^{-1}$, and $\nu_{3}\;=\;1801$ cm${}^{-1}$ from their best QFF. These results differ quite significantly from previous results [32] by as much as 9 cm${}^{-1}$ for the fundamental vibrational transtions and 0.0014 Å for the CH bond length, the latter thus well outside the estimated error bar of Mladenović and coworkers. Finally, Morgan and Fortenberry [34] also investigated HCC${}^{-}$ by use of a similar composite QFF scheme [43]. However, their results are somewhat more in line with the work of Mladenović et al. given that no correction for higher-order correlation based on ACPF has been included.

It appears that some of the theoretical results are somewhat questionable. This could be problematic for future spectroscopic studies on HCC${}^{-}$ since comparison between experiment and theory might not be conclusive. Therefore, this study aims at providing benchmark quality predictions for the rovibrational spectrum of the acetylide anion. To this end, a high-level ab initio composite approach is employed in the construction of a PES. Variational calculations employing the composite PES then provide vibrational band origins with an accuracy of around 1 cm${}^{-1}$. Combined with an electric dipole moment surface (EDMS) the rovibrational spectrum up to about the first excited CH stretching state ∼3300 cm${}^{-1}$ will be investigated. Finally, comparisons with published composite approaches to HCC${}^{-}$ will highlight shortcomings of the previous work which should prove valuable for future theoretical investigations not only of the title compound.

## 2. Methods

### 2.1. Electronic Structure Calculations

The construction of the PES is based on a well established composite procedure which has been applied to a variety of small polyatomic molecules [44,45,46,47,48,49,50,51,52,53]. The employed contributions closely follow the Feller-Peterson-Dixon partitioning of the energy [54,55,56,57,58,59,60]. Here, the basic contribution is obtained from frozen-core (fc) explicitly correlated coupled-cluster calculations with singles, doubles and (scaled) perturbative triples CCSD(T*)-F12b [61,62,63] together with an aug-cc-pV5Z basis set [64,65]. This level of theory will be abbreviated as F12bs in the following. The auxiliary basis sets, as required for F12-calculations, were AV5Z/OPTRI [66], AV5Z/JKFIT [67], and AV5Z/MP2FIT [68]. As suggested by Peterson et al. [69] a geminal parameter β of 1.5 $a_{0}^{-1}$ has been used.

The composite procedure adds small corrections to the basic F12bs contribution to incorporate core-core and core-valence correlation (CV), scalar-relativisitic effects (SR), higher-order correlation (HC) beyond CCSD(T), and the diagonal Born-Oppenheimer-Correction (DBOC). In detail these effects are calculated in the following ways:**CV:** The CV effects are captured by conventional CCSD(T) calculation and a large (781 contracted Gaussian-type orbitals) aug-cc-pCV6Z basis set [70]. By subtracting the result of frozen-core CCSD(T) from that of an all-electron (ae) calculation the CV contribution is obtained.**SR:** Second-order Dogulas-Kroll-Hess calculations [71,72] are employed at the fc-CCSD(T) level to provide SR effects, where difference of calculations with the relativistic Hamiltonian employing a aug-cc-pVQZ-DK basis set [73] and non-relativistic calculations with an aug-cc-pVQZ basis [64,65] yields the contribution to the composite PES.**HC:** Three distinct HC contributions are employed. The first, which is termed (Q)–(T), corrects for the fully iterative treatment of triple excitations as well as for a quadruple excitation effect by a perturbative treatment. The second contribution Q–(Q) accounts for the full iterative treatment of quadruples. This partitioning has been shown to be rather efficient since larger basis sets can be used for (Q)–(T) compared to Q–(Q). Finally, the P–Q contribution incorporates pentuple excitation contributions.
**(Q)–(T):** This contribution employs the fc-CCSDT(Q) method [74] and is obtained as the difference to fc-CCSD(T). The calculations are performed with an aug-cc-pVTZ basis set [64].**Q–(Q):** The second HC contribution is obtained by forming the difference of fc-CCSDTQ [75] with respect to fc-CCSDT(Q). The basis is constructed by omiting the highest angular momentum functions in the cc-pVTZ basis [64], i.e., *f*-functions are omitted for carbon and *d*-functions for hydrogen.**P–Q:** The third HC contribution is obtained from the difference of fc-CCSDTQP [75] and fc-CCSDTQ together with the cc-pVDZ basis [64].Karton and coworkers demonstrated repeatedly [76,77,78,79] that basis set dependencies of HC effects diminishes with the excitation level.**DBOC:** All-electron CCSD calculations are employed for the adiabatic DBOC correction [80,81,82] together with a cc-pCVQZ basis [70]. The DBOC calculations were perfomed with the Cfour program [83,84]. Due to the inverse mass dependence this contribution is important for hydrogen containing systems. It is evaluated for 6 possible H/D and ${}^{12}$C/${}^{13}$C combinations, i.e., H/D substitution and single ${}^{12}$C/${}^{13}$C switch, and in turn the inclusion of DBOC leads to 6 different adiabatic PESs for HCC${}^{-}$ isotopologues.

All electronic structure calculations, unless noted otherwise, have been performed with the Molpro suite of ab initio programs [85,86,87] and an interface to Mrcc by Kállay and coworkers [88,89] for the HC calculations.

The linear reference structure employed for all contributions corresponds to the recommended bond lengths by Botschwina and coworkers [32], i.e., a CH bond length of $r_{\mathrm{ref}}\;=\;1.0689$ Å, and a CC bond length of $R_{\mathrm{ref}}=1.2464$ Å. The PES is sampled in the ranges −0.175Å≤Δr≤0.250Å, −0.150Å≤ΔR≤0.200Å, and $0^{\circ }\leq \theta \leq 75^{\circ }$, where the internal coordinates Δr and ΔR are simple bond stretches and the angle θ measures the deviation from linearity. These ranges cover energies up to ∼10,000 cm${}^{-1}$ above the reference along the diagonal cuts of the PES. Each contribution (labeled α) is represented by a least-squares fit to a polynomial form
(1)$$V^{\left(\alpha \right)}-V_{\mathrm{ref}}^{\left(\alpha \right)}\;=\;\sum_{ijk}C_{ijk}^{\left(\alpha \right)}\;\Delta r^{i}\;\Delta R^{j}\;\theta^{k}\;.$$

In Equation (Equation 1) the exponent *k* is even as required by symmetry. The individual fitted contributions are summed up and transformed to the minimum. This procedure yields composite PESs and equilibrium bond lengths $r_{e}$ and $R_{e}$.

The EDMS is developed by combining the F12bs, CV, SR, and HC contributions employed for the PES. This allows the calculation of transition dipole moments and rovibrational line intensities. The EDMS is sampled in the same coordinate range as the PES and the dipole moment vector is obtained by finite field calculations (±0.0003 a.u.), i.e., for the difference based contributions this involves up to 8 individual calculations. Molecule fixed components of the dipole vector (parallel $\mu_{\|}$ and perpendicular $\mu_{⊥}$) are obtained by locally transforming to the Eckart frame of the main isotopologue HCC${}^{-}$. Each component of the individual contributions (α) is then least-squares fit to a polynomial form:
(2a)$$\mu_{\|}^{\left(\alpha \right)}\;=\;\sum_{ijk}D_{ijk}^{\left(\alpha \right)}\;\Delta r^{i}\;\Delta R^{j}\;\theta^{k}\;(k:\;\mathrm{even})$$
(2b)$$\mu_{⊥}^{\left(\alpha \right)}\;=\;\sum_{ijk}D_{ijk}^{\left(\alpha \right)}\;\Delta r^{i}\;\Delta R^{j}\;\theta^{k}\;(k:\;\mathrm{odd})\;.$$

The EDMS of other isotopologues is obtained by transforming the components to the new Eckart frame during the rovibrational calculations.

### 2.2. Rovibrational Calculations

The acetylide anion is a rather semi-rigid system in the lower spectral region [32]. Therefore, rovibrational calculations based on normal coordinates and the Eckart-Watson isomorphic Hamiltonian [90,91,92,93] are feasible. The two common approaches are based on either second-order vibrational perturbation theory (VPT2) [94,95,96,97] or variational calculations using harmonic oscillator/rigid rotor product basis functions [98,99,100]. One-dimensional harmonic oscillators [101] are used for the two stretching coordinates and 2D isotropic harmonic oscillator functions for the degenerate bending [102]. Both approaches either based on perturbation theory or on the variational principle will be employed and compared in the present work. All (ro)vibrational calculations employ atomic masses [103].

The appeal of VPT2 is that this treatment leads to compact formula for spectroscopic parameters [104,105] and in absence of resonance interactions is easy to apply. To this end, the composite PESs are transformed to a QFF representation
(3)$$V\;=\;\frac{1}{2}\sum_{i}\omega_{i}q_{i}^{2}+\frac{1}{6}\sum_{ijk}\phi_{ijk}q_{i}q_{j}q_{k}+\frac{1}{24}\sum_{ijkl}\phi_{ijkl}q_{i}q_{j}q_{k}q_{l}\;.$$

In Equation (Equation 3), $\omega_{i}$ is the harmonic vibrational frequency. The cubic and quartic force constants are denoted by $\phi_{ijk}$ and $\phi_{ijkl}$, respectively. They are obtained by numerical differentiation of the composite PES with respect to dimensionless normal coordinates $q_{r}$. Various (ro)vibrational spectroscopic parameters can then be calculated by standard formulae [104,105,106,107,108], e.g., anharmonicity constants $x_{ij}$ defining vibrational term energies for triatomic linear molecules
(4)$$\begin{array}{cc} G_{v}\;=\;E_{0} & +\sum_{i}\omega_{i}\left(v_{i}+\frac{d_{i}}{2}\right)+\sum_{i<j}x_{ij}\left(v_{i}+\frac{d_{i}}{2}\right)\left(v_{j}+\frac{d_{j}}{2}\right)+x_{ll}l^{2} \end{array}$$
or rotation-vibration coupling constants $\alpha_{i}$ which provide the vibrational dependencies of rotational constants according to
(5)$$B_{v}\;=\;B_{e}-\sum_{i}\alpha_{i}\left(v_{i}+\frac{d_{i}}{2}\right)\;.$$

In Equations (Equation 4) and (Equation 5) $E_{0}$ is a constant (quantum number $v_{r}$ independent) contribution to the zero-point vibrational energy (ZPVE) [105,106,107,108], $d_{r}$ the degeneracy of the respective vibrational mode and $B_{e}$ the equilibrium rotational constant. The vibrational angular momentum quantum number for the degenerate bending mode is given by *ℓ*.

The variational calculations are performed with Sebald’s C8vpro program [109] to obtain rovibrational term energies $T_{v}\left(J\right)$ and wavefunctions. To this end, the Hamiltonian up to a total angular momentum quantum number of $J_{\max}=60$ is diagonalized in a basis of 2360 Wang-symmetrized [110,111] basis functions per *K*-block, where *K* is the quantum number for the projection of the total angular momentum unto the molecular axis. For linear triatomic systems the Sayvetz rule [91] requires K=l. In order to reduce the computational costs the maximum value of *ℓ* was fixed at 8. This “frozen *ℓ*” approach has been succesfully applied previously even to floppy C${}_{3}$ [52]. In summary, this basis yields vibrational term energies converged to within 0.1 cm${}^{-1}$ or better for the states of interest.

In the following, vibrational states of HCC${}^{-}$ are designated by *v* which collects the label $\left(v_{1},v_{2}^{l},v_{3}\right)$ with $v_{1}$, $v_{2}$, and $v_{3}$ being the vibrational quantum numbers of the CH-stretching, bending, and CC-stretching, respectively. For states with l=0 spectroscopic parameters can be determined by least-squares fitting the variational term energies according to
(6)$$T_{v}\left(J\right)\;=\;G_{v}+B_{v}J\left(J+1\right)-D_{v}J^{2}{\left(J+1\right)}^{2}+H_{v}J^{3}{\left(J+1\right)}^{3}\;,$$
where $G_{v}$ is the vibrational term energy and $B_{v}$ is the rotational constant. The quartic and sextic centrifugal distortion parameters are given by $D_{v}$ and $H_{v}$, respectively. Effects due to *ℓ*-type doubling and resonance [112,113,114] in states with l≠0 are accounted for by setting up effective Hamiltonian matrices in conjunction with least-squares fitting to the variational rovibrational term energies. The diagonal and off-diagonal elements are given by [96]
(7)$$\begin{array}{cc} \langle v,J,l\;|\;{\hat{H}}_{\mathrm{eff}}\;|\;v,J,l\rangle \;=\;G_{v}+B_{v}\left[J\left(J+1\right)-l^{2}\right] & -D_{v}{\left[J\left(J+1\right)-l^{2}\right]}^{2}\\ & +H_{v}{\left[J\left(J+1\right)-l^{2}\right]}^{3} \end{array}$$
and
(8)$$\begin{array}{cc} \langle v,J,l\;|\;{\hat{H}}_{\mathrm{eff}}\;|\;v,J,l\pm 2\rangle \;=\; & \frac{1}{4}\left[q_{v}+q_{v}^{J}J\left(J+1\right)+q_{v}^{JJ}J^{2}{\left(J+1\right)}^{2}+\ldots \right]\\ \; & \times \left\{\left[{\left(v_{2}^{2}+1\right)}^{2}-{\left(l\pm 1\right)}^{2}\right]\left[J(J+1)-l(l\pm 1)\right]\right.\\ \; & \times {\left.\left[J(J+1)-(l\pm 1)(l\pm 2)\right]\right\}}^{1/2}\;, \end{array}$$
respectively. In Equation (Equation 8) $q_{v}$, $q_{v}^{J}$, and $q_{v}^{JJ}$ are the *ℓ*-type doubling constants.

Line intensities $A_{\mathrm{if}}$ for rovibrational transitions at a temperatur *T* between initial states with energy $E_{i}$ and final states with energy $E_{f}$ are given by
(9)$$A_{\mathrm{if}}\;=\;\frac{8\pi^{3}}{3hc}\;g\;\frac{e^{-E_{i}/k_{B}T}\cdot \left[1-e^{-\left(E_{f}-E_{i}\right)/k_{B}T}\right]}{Q(T)}\;\left(E_{f}-E_{i}\right)\;\mu_{\mathrm{if}}^{2}\;.$$

In Equation (Equation 9), the first factor contains Planck’s constant *h* as well as the speed of light *c*. Boltzmann’s constant is denoted by $k_{B}$. Furthermore, *g* is the statistical weight (accounting for isotopical abundance), $\mu_{\mathrm{if}}^{2}$ the squared transition dipole moment and Q(T) the total internal partition function.

## 3. Results and Discussion

### 3.1. Construction of the PES and EDMS

As a first step in the development of the composite PES, the basis set convergence of the equilibrium bond lengths and and harmonic vibrational frequencies is investigated. Conventional CCSD(T) result with very large basis sets up to AV8Z are provided in Table 1. The AV7Z and AV8Z calculations have been perfomed with Dalton [115,116]. Results for explicitly correlated CCSD(T)-F12b and the triples scaled variant CCSD(T*)-F12b, i.e., the F12bs contribution, are provided as well. The CCSD(T) equilibrium bond lengths show a systematic convergence and the AV8Z are probably within about 0.00005 Å of the CBS limit. That is, no further extrapolation appears to be neccessary. The CCSD(T)-F12b results appear to slightly overshoot the AV8Z calculations. This is remedied by the triples scaling. Such behaviour has been observed previously for similar triatomic systems when using augmented basis sets [52].

The convergence of the harmonic vibrational frequencies is also graphically displayed in Figure 1. The F12bs results clearly are very close to the CBS limit and somewhat more consistent with the conventional CCSD(T) values than without triples scaling. In summary F12bs provides near-CBS quality results at a significantly reduced computational cost which is important when sampling a PES.

The dependence of smaller contributions on the internal coordinates is presented in Figure 2. The dominant contribution for the stretching coordinates Δr (upper panel) and ΔR (lower panel) are CV effects. Effects due to HC only show a slight dependence on the CH-stretch whereas they are still significant for ΔR and act in the opposite direction of the CV contribution. These trends are in line with what has been observed previously [44,45,46,47,48,49,50,51,52,53]. In case of the bending motion (inset lower panel) it is clear that the inclusion of (Q)–(T) is important with a contribution of about −58 cm${}^{-1}$ at $60^{\circ }$. For comparison the isoelectronic HCS${}^{+}$ ion [51] yields a slightly larger value of −67 cm${}^{-1}$.

The impact of the smaller contributions on the equilibrium bond lengths and harmonic vibrational frequencies is presented in Table 2.

While CV effects dominate the shifts in $r_{e}$ and $\omega_{1}$, the combination of CV and HC is necessary for the CC-stretching and most importantly for the bending, in line with Figure 2. The total HC correction, i.e., the sum of the present three individual effects, amounts to +0.00007 and +0.00080 Å for $r_{e}$ and $R_{e}$, respectively, as well as −2.8, −6.0, and −6.0 cm${}^{-1}$ for $\omega_{1}$, $\omega_{2}$, and $\omega_{3}$, respectively. Comparing the full composite results to the basic F12bs the overall difference is significant for the bond lengths with changes of about −0.0005 and −0.0028 Å. In contrast, for the harmonic vibrational frequencies there is some amount of compensation between the different contributions, e.g., for the bending vibration $\omega_{2}$ F12bs and the composite PES differ only by about 1 cm${}^{-1}$ and doesn’t exceed ∼6 cm${}^{-1}$ for stretches. Nevertheless, when aiming at high-accuracy their inclusion is of course mandatory.

It is interesting to compare the composite results to those published earlier [32,33,34] (cf. lower part of Table 2). The results provided by Mladenović et al. are very close despite the fact that their employed theoretical level is only approximately ae-CCSD(T)/ACVQZ. While this is not surprising for the bond lengths which include a further empirical correction based on equivalent calculations for HCN, the harmonic vibrational frequencies provided by Mladenović et al. are within less than 1 cm${}^{-1}$ compared to the present ones. Clearly this is due to fortuitous error compensation between the basis set incompleteness with ACVQZ and missing HC effects. The incompleteness error can be estimated by comparing the ae-CCSD(T)/ACVQZ harmonic frequencies with composite F12bs + CV, the latter of which should provide close to CBS limit. This yields values of −2.4, −6.1, and −5.8 cm${}^{-1}$ in almost perfect agreement with the shifts due to the summed HC effects given above. The results of Morgan and Fortenberry [34] can also be compared to the F12bs + CV + SR. At that level of theory the bond lengths are $r_{e}=1.06945$ and $R_{e}=1.24708$ Å which are larger than the F12bs + CV + SR results by 0.00046 and 0.00168 Å, respectively. Given the formally comparable level of theory this large difference is surprising. Moreover, while both methods yield almost identical results for $\omega_{2}$ and $\omega_{3}$ the CH-stretching harmonic vibrational frequency $\omega_{1}$ differs by 1.9 cm${}^{-1}$. This may suggest some numerical instabilities in the results of Morgan and Fortenberry since large differences in a bond length usually are associated with changes in the harmonic frequncies that are associated with that bond, i.e., the significant difference of 0.00168 Å in the CC bond length would suggest a larger deviation in $\omega_{3}$ the CC stretching which, however, is not observed.

In contrast to the good agreement with the comparatively lower level results of Mladenović et al., the results of Huang and Lee [33] show very large differences despite the fact that they account for similar corrections. Compared to the present composite PES, their $r_{e}$ and $R_{e}$ equilibrium bond lengths differ by as much as 0.00172 and 0.00081 Å, respectively and harmonic frequencies by 1.8, 16.5, and 6.0 cm${}^{-1}$ for $\omega_{1}$, $\omega_{2}$ and $\omega_{3}$, respectively. As discussed in the introduction Huang and Lee constructed a composite QFF combining CBS extrapolated CCSD(T) results with smaller correction due to CV, SR, and HC effects. The latter were obtained as the difference between CBS-extrapolated ACPF and CCSD(T) calculations (termed 3-pt AC/AVXZ in Ref. [33]). A closer look at Table III in Ref. [33] shows that this way of approximating HC appears to overestimate the effects. For example the harmonic frequencies are shifted by +3.7, −21.4 and −12.6 cm${}^{-1}$ for $\omega_{1}$, $\omega_{2}$ and $\omega_{3}$, respectively. These numbers are significantly different than the present HC effects by factors of about 1.5, 3.5 and 2.2, including a sign change for the CH-stretching contribution.

In order to rule out numerical issues due to the fitting procedure, graphical representations for 1D scans of the ACPF−CCSD(T) difference (including the same 3-pt CBS extrapolation as Huang and Lee [33,117]) along the internal coordinates Δr, ΔR, and θ are provided in Figure 3. Furthermore, to check wether the problems are due to the treatment of dynamical correlation Figure 3 also shows results obtaind with multi-reference configuration interaction with singles and doubles [118,119,120,121,122] including the Davidson correction (MRCI+D) [123,124,125] and averaged quadratic coupled-cluster (AQCC) [126]. Finally, results obtained with explicitly correlated methods [127,128] and an AV5Z basis are also shown. From inspection of Figure 3 it is clear that, while there are subtle differences between ACPF, MRCI+D, and AQCC, none of the methods provides an accurate description of HC effects beyound CCSD(T). These difficulties could indicate that the actual problem is within the the reference used for the multi-reference approaches. It is well known that full-valence CASSCF can lead to problems in providing an appropriate active space. The reader is referred to the works by Veryazov et al. [129] as well as Stein and Reiher [130] for general discussions of this problem. Furthermore, the works of Makhnev et al. [131] on isoelectronic HCN and Schröder et al. [50] on nitrous oxide (N${}_{2}$O) have highlighted this issue when constructing high-level ab initio PESs for triatomic systems. In summary, the HC contribution employed by Huang and Lee [33] based on the ACPF-CCSD(T) differenc is not recommended.

The final composite PES is constructed from 165 symmetry unique points for F12bs whereas a lower number of points—especially for the 2D and 3D coupling portion of the PES—is required to accurately determine the smaller contributions. These data points are available at GRO.data [132]. The least-squares fit according to Equation (Equation 1) employs up to powers of 8 and 10 for the diagonal stretching and bending monomials, respectively. For the 2D and 3D couping terms a total polynomial degree of 6 is found to be sufficient in the target energy regime. The residual error of the F12bs least-squares fit is 0.008 cm${}^{-1}$ and even smaller for the individual contributions. Appendix A provides the coefficients $C_{ijk}^{\left(\alpha \right)}$. The polynomial representations of the adiabatic PESs can be found in Appendix A.

Table 3 presents fc-CCSD(T) and fc-CCSD(T)-F12b results for the equilibrium dipole moment $\mu_{e}$ and band intensities of fundamental vibrational transitions $A_{0i}$ for the main isotopologue. The negative sign for the dipole moment $\mu_{e}$ indicates the direction of the dipole vector pointing from the terminal C to the H atom, i.e., a polarity according to +HCC−. Band intensities are obtained within the double harmonic approximation, i.e., harmonic wave function and linear dependency of the dipole moment with respect to normal coordinates. Using the present composite PES defines the harmonic wave function for the squared transition dipole moments. Both $\mu_{e}$ and the $A_{0i}$ appear to be converged to within about 0.001 D and 0.2 km/mol or better, respectively, at the fc-CCSD(T)/AV6Z level of theory. The standard CCSD(T)-F12b and its triples scaled variant are very close to each other and match the conventional AV6Z results very well. An interesting observation regarding the stretching fundamentals can be made. While in isoelectronic HCN the CN-stretching band is very weak compared to the CH-stretching [133,134], the situation is reversed for HCC${}^{-}$. In both systems the lower of the two streching band intensities results from a compensation effect of the dipole moment derivatives with respect to the (internal) stretching coordinates upon transformation to normal coordinates.

In Table 4 the numerical influence of the smaller contributions is given. All effects are rather small in absolute numbers which has also been observed previously by Schröder et al. [50] for N${}_{2}$O. However, due to the small band intensity of the CH-stretching fundamental the relative change from F12bs (1.77 km/mol) to the composite result (2.27 km/mol) is almost 30%. The effect on the other band intensities is significantly lower. The final EDMS is determined from the same 165 points as employed for the PES. Least-squares fits according to Equation (2) were perfomed with terms up to a total polynomial degree of 5 yielding fitting residuals on the order of about 10${}^{-5}$ D for the individual contributions.

The coefficients $D_{ijk}^{\left(\alpha \right)}$ are provided in Appendix A and the combined polynomial representation of the HCC${}^{-}$ EDMS can be found in Appendix A.

### 3.2. Spectroscopic Parameters from VPT2

VPT2 allows a rather compact assessment of the influence of anharmonicity on the rovibrational spectrum due to straightforward formulae to calculate spectroscopic parameters. In turn, using VPT2 allows to take a more detailed look at the influence of smaller contributions. Table 5 presents results for the main isotopologue of HCC${}^{-}$.

The commonly reported VPT2 based spectroscopic parameters and their change upon inclusion of the different effects are quoted. It is clear that the observation made for harmonic frequencies, i.e., a large degree of compensation between the contributions, is more pronounced for anharmonicity constants $x_{ij}$ and subsequently for anharmonic corrections to vibrational transtion frequencies $\Delta_{i}$. However, for highly accurate rotational constants the inclusion of higher-order correlation appears to be mandatory. Overall the relative difference between the full composite PES and F12bs amounts to less than 1% on average, with the exception of $x_{22}$ which due to the very small value shows a large relative shift of more than 50%.

### 3.3. Results of Variational Calculations

The C8vpro outputs containing the lowest 90 variationally determined eigenvalues of the rovibrational Hamiltonian up to $J_{\max}=60$ for HCC${}^{-}$ isotopologues are available from GRO.data [132]. In the following spectroscopic parameters and spectra determined from these calculations will be discussed. The only experimental spectroscopic information available for HCC${}^{-}$ has been obtained from rotational spectroscopy. Therefore the vibrational ground state is investigated first and Table 6 presents the relevant spectroscopic parameters for selected HCC${}^{-}$ isotopologues. For comparison the results of Brünken and coworkers [16] as well as Amano [26] are provided. The present theoretical rotational constants $B_{0}$ obtained from variational calculations agree to within about 0.005% with their experimental counterparts. Interestingly, the $B_{0}$ obtained from VPT2 of 41,637.8 MHz is even slightly closer to experiment. This changes when looking at the centrifugal distortion parameters. The variational results of 96.871, 92.581, and 90.080 kHz for H^12^C^12^C${}^{-}$, H^13^C^12^C${}^{-}$ and, H^12^C^13^C${}^{-}$, deviate from experiment by not more than 0.1%.

In contrast, VPT2 yields a value of 94.1 kHz for H^12^C^12^C${}^{-}$, i.e., a deviation by 2.9% can be observed. As is well known, VPT2 based quartic centrifugal distortion constants $D_{e}$ for linear molecules do not include contributions from bending vibrations and lack effects due to vibrational averaging [45,135,136,137]. In their determination of the rotational spectroscopic parameters both experimental works [16,26] fixed $H_{0}$ to a value recommended by Sebald and Botschwina based on the PES reported earlier [32]. Clearly this value is a reasonable choice as the present calculations yield almost the same value based on a much more accurate PES. Finally, the ZPVE of 3078.0 cm${}^{-1}$ calculated variationally for H^12^C^12^C${}^{-}$ is in perfect agreement with the VPT2 result as is expected for such a semi-rigid molecule.

It may appear tempting to use the results for H${}^{12}$C${}^{12}$C${}^{-}$, H${}^{13}$C${}^{12}$C${}^{-}$, and H${}^{12}$C${}^{13}$C${}^{-}$ to deduce a semi-experimental equilibrium structure. This is a well established approach [138,139,140,141,142,143,144,145] where the theoretically calculated difference $\Delta B_{0}=B_{e}-B_{0}$ is used to correct experimental ground state rotational parameters to obtain equilibrium values which then can be converted to an equilibrium geometry. However, the absence of data for deuterated isotopologues will make the determination of a semi-experimental $r_{e}$ questionable. The outlined approach yields values of $r_{e}=1.06951$ Å and $R_{e}=1.24618$ Å upon combination of the $B_{0}$ from experiment [25,26] with $\Delta B_{0}$ obtained from variational calculations. While the latter value is in perfect agreement with the ab initio result of 1.24621 Å, the difference of 0.00034 Å obtained for the $r_{e}$ is significantly larger than what has been observed previously [50,52]. Note that this way of calculating semi-experimental equilibrium structures can be used even for systems with strong rovibrational coupling yielding close agreement to high-level composite ab initio results (see, e.g., Ref. [52]). Nevertheless, considering the close agreement to the $B_{0}$ from experiment an updated recommended linear equilibrium structure with $r_{e}\;=\;1.0692\left(2\right)$ Å and $R_{e}=1.2462\left(2\right)$ Å appears to be appropriate, where a conservative error estimate in terms of the last significant digit is given in parentheses.

Calculated spectroscopic parameters for the singly excited states $\left(1,0^{0},0\right)$, $\left(0,1^{1},0\right)$, and $\left(0,0^{0},1\right)$ are collected in Table 7. No experimental information is yet available for comparison and therefore these results stand as predictions. Spectroscopic parameters for additional selected rovibrational states in H^12^C^12^C${}^{-}$ are provided in Appendix A. Based on previous experience with similar molecules [46,47,49,50,51,52] the vibrational term energies $G_{v}$ are expected to be accurate to within 1 cm${}^{-1}$ and rotational parameters should display the same level of accuracy as observed for the vibrational ground state. Comparing the variational results for $G_{v}$ in H^12^C^12^C${}^{-}$ with the corresponding composite VPT2 values for $\nu_{i}$ in Table 5 almost perfect agreement can be observed as expected for a semi-rigid molecule.

The fortuitous error compensation mentioned earlier for the results of Mladenović et al. [32] can also be extended to the anharmonic vibrational term energies. The latter authors obtained $G_{v}$ values (in cm${}^{-1}$) of 511.1, 1805.0, and 3211.3 for $\left(0,1^{1},0\right)$, $\left(0,0^{0},1\right)$, and $\left(1,0^{0},0\right)$, respectively, with an ae-CCSD(T) based PES to be compared with the present composite results of 510.1, 1804.5, and 3209.6. In contrast, the rotational parameters calculated by Mladenović et al. [32] are less accurate than the present results which can be traced back to differences in the equilibrium geometry used by these authors in their rovibrational calculations ($r_{e}\;=\;1.0697$ Å and $R_{e}\;=\;1.2474$ Å).

The $\left(1,0^{0},0\right)$ state of the deuterated isotopologues shows signs of an anharmonic resonance. This can be inferred from the anomalous large values of $H_{v}$ and by comparing the effective $\alpha_{1}\;=\;B_{0}-B_{1}$ with VPT2 results. The latter values (in MHz) are calculated from the variational calculations to be 284.6, 268.9, and 206.3, for D^12^C^12^C${}^{-}$, D^13^C^12^C${}^{-}$, and D^12^C^13^C${}^{-}$, respectively, to be compared with the VPT2 results of 301.0, 282.2, and 286.2. Note that for H^12^C^12^C${}^{-}$ this analysis yields 299.0 MHz and 295.7 from variational calculations and VPT2, respectively, in good agreement. Upon inspection of the rovibrational wave functions the $\left(0,2^{0},1\right)$ state is identified as the perturbing state. The energetic situation is graphically depicted in Figure 4. Such a 1-3 Darling-Dennison resonance [107,146,147] can be analyzed in terms of the vibrational term energies by setting up an effective Hamiltonian of the form
(10)$${\underline{\underline{H}}}_{\mathrm{eff}}=\left(\begin{array}{cc} G_{1,0^{0},0}^{*} & K_{\mathrm{DD}}\\ K_{\mathrm{DD}} & G_{0,2^{0},1}^{*} \end{array}\right)\;.$$

In Equation (Equation 10), $G_{v}^{*}$ are so-called “deperturbed” vibrational term energies and $K_{\mathrm{DD}}$ is the Darling-Dennison coupling matrix element. From two perturbed J=0 term energies of the $\left(1,0^{0},0\right)$ and $\left(0,2^{0},1\right)$ states alone one can not determine the three parameters in the effective Hamiltonian of Equation (Equation 10). However, given the excellent agreement between VPT2 and the variational calculations observed for HCC${}^{-}$ one can fix the $G_{1,0^{0},0}^{*}$ to the perturbational results 2478.2, 2449.9, and 2473.4 cm${}^{-1}$ for D^12^C^12^C${}^{-}$, D^13^C^12^C${}^{-}$, D^12^C^13^C${}^{-}$, respectively. This yields $G_{0,2^{0},1}^{*}$ values of 2515.7, 2488.0, and 2481.5 cm${}^{-1}$, to be compared with the VPT2 results of 2515.5, 2487.6, and 2481.2, for D^12^C^12^C${}^{-}$, D^13^C^12^C${}^{-}$, D^12^C^13^C${}^{-}$, respectively, showing good agreement. The coupling constants differ only slightly between the isotopologues (cf. also Figure 4) with an average value of 6.6 cm${}^{-1}$. For a full analysis, i.e., for a deperturbation of rotational parameters, the effective Hamiltonian from Equation (Equation 10) would need to be extended to allow for the simultaneous *ℓ*-type resonance between $\left(0,2^{0},1\right)$ and the $\left(0,2^{2},1\right)$ state with *e*-parity [51]. However, no attempt in that direction has been undertaken here and therefore the parameters for DCC${}^{-}$ isotopologues have to be treated as effective parameters.

The ground state dipole moment $\mu_{0}$ of HCC${}^{-}$ is calculated to be −3.100 D very close to the earlier results of Mladenović et al. [32] Comparing $\mu_{0}$ and $\mu_{e}$ from Table 4 one finds that the vibrational averaging reduces the absolute value by 0.121 D or 4 %. The latter quantity can also be evaluated through 2nd order via the differences $\Delta \mu_{i}\;=\;\mu_{i}-\mu_{0}$. Using the dipole moments $\mu_{i}$ from the variational calculations of the vibrational states $\left(1,0^{0},0\right)$ ($\mu_{1}\;=\;-$3.054 D), $\left(0,1^{1},0\right)$ ($\mu_{2}\;=\;-$3.014 D), and $\left(0,0^{0},1\right)$ ($\mu_{3}\;=\;-$3.073 D), one arrives at 0.096 D in good agreement. The remaining difference is probably due to higher-order contributions. In contrast, H/D substitution changes $\mu_{0}$ by 11 % (DCC${}^{-}$$\mu_{0}\;=\;-$3.431 D) which is predominantly due to the change in the equilibrium dipole moment which has to be considered for charged species. Similar results have been obtained previously for isoelecronic HCS${}^{+}$ by Schröder and Sebald [51].

Finally, rovibrational line intensities for the fundamental transitions were obtained following Equation (Equation 9). At T=300 K the total internal partition function Q(T) amounts to 180.33 for the main isotopologue, based on the present rovibrational calculations. Results for other isotopologues as well as different temperatures are collected in the Appendix A. Figure 5 compares stick spectra for the fundamental transtions in HCC${}^{-}$ and DCC${}^{-}$. Note that *g* in Equation (Equation 9) has been set to 1 to allow for direct comparison. The $\nu_{2}$ band is found to be the most intense fundamental transition followed by the CC-stretching $\nu_{3}$ which is weaker by a factor of about 2. These results are in line with the spectra calculated by Mladenovic et al. [32]. In all cases exchanging H for D reduces the intensities.

An interesting effect can be observed for the $\nu_{1}$ band, where upon H/D substitution the intensity ratio of the P- and R-branch reverses. Note that Mladenovic et al. [32] predicted this reversed intensity ratio also for the main isotopologue (cf. Figure 2 of Ref. [32]) but the present calculations based upon a highly accurate PES and EDMS do not reproduce this observation. To understand the intensity pattern a closer look at the squared transition dipole moment $\mu_{\mathrm{if}}^{2}$ is necessary. This quantity can be approximated according to [148]
(11)$$\mu_{\mathrm{if}}^{2}\approx \mu_{vv^{\prime }}^{2}F_{\mathrm{HL}}F_{\mathrm{HW}}\;,$$
where $\mu_{vv^{\prime }}$ is the vibrational transition moment, $F_{\mathrm{HL}}$ the Hönl-London factor [149,150], and $F_{\mathrm{HW}}$ the Herman-Wallis factor [151]. The latter quantity can be written as
(12)$$F_{\mathrm{HW}}\;=\;{\left(1+A_{1}m+A_{2}m^{2}\right)}^{2}\;,$$
where *m* is −J or J+1 for P- and R-branch transitions, respectively, and according to Watson [152] the $A_{1}$ coefficient for a parallel fundamental band $\nu_{s}$ within VPT2 is found to be
(13)$$A_{1}\;=\;-2\frac{B_{s}}{\omega_{s}}\frac{\mu_{e}}{\mu_{s}}+4B_{e}\sum_{t}\zeta_{st}\frac{\sqrt{\omega_{s}\omega_{t}}}{\omega_{s}^{2}-\omega_{t}^{2}}\frac{\mu_{t}}{\mu_{s}}\;.$$

In Equation (Equation 13), indices *s* and *t* label stretching and degenerate bending normal modes, respectively. The two terms correspond to two effects responsible for the *m*-dependence of the effective transition moment; while the first term describes mixing with the rotational spectrum, the second term is due to Coriolis coupling of the stretching with the bending fundamental band. The $B_{s}$ are rotational derivatives [97], $\zeta_{st}$ the Coriolis coupling constants and $\mu_{k}$ the first derivative of the dipole moment with respect to the dimensionless normal coordinate $q_{k}$. With the present PES and EDMS one obtains 0.012 and −0.023 for $A_{1}$ coefficient of the $\nu_{1}$ band in HCC${}^{-}$ and DCC${}^{-}$, respectively. From the variationally calculated $\mu_{\mathrm{if}}^{2}$ application of Equations (Equation 11) and (Equation 12) also allows the determination of $A_{1}$ through least-squares fitting resulting in slightly smaller values of 0.0072 and −0.0072, in the previous ordering of isotopologues. The change of sign is responsible for the inverted P/R intensity ratio and upon inspection of the contributions to $A_{1}$ in Equation (Equation 13) is due to a change of sign of the dipole moment derivative $\mu_{1}$ upon H/D substitution.

It is clear that the most promising target for the detection of HCC${}^{-}$ will be the $\nu_{2}$ bending fundamental. Therefore the spectrum at T=300 K in the relevant range is presented in Figure 6. The spectrum includes additional hot bands, i.e., $\left(0,2^{l},0\right)\leftarrow \left(0,1^{1},0\right)$ and $\left(0,3^{l},0\right)\leftarrow \left(0,2^{l},0\right)$ as well as the same bands for all singly ${}^{13}$C substituted isotopologues in natural abundance. The spectra of H^13^C^12^C${}^{-}$ and H^12^C^13^C${}^{-}$ are weaker by a factor of 100 in agreement with the reduced abundance of ${}^{13}$C. Both Q-branches of the $\nu_{2}$ bands in the ${}^{13}$C isotopologues are located inbetween the P- and Q-branch of H^12^C^12^C${}^{-}$ and thus might be possible to detect. Hot bands of H^12^C^12^C${}^{-}$ have slightly larger intensity compared to the other isotopologue bands and should therefor also be observable.

In order to facilitate future experimental work, Appendix A collects band intensities for a number of overtones, combination bands as well as hot bands in H^12^C^12^C${}^{-}$. These have been obtained by summing over individual rovibrational lines within a given vibrational transition.

## 4. Summary

The present work provided a thorough theoretical invesitgation into the rovibrational spectrum of the acetylide anion (HCC${}^{-}$) and its isotopologues. An ab initio potential energy surface (PES) and electric dipole moment surface (EDMS) was constructed by a high-level composite procedure within the framework of coupled-cluster theory. Explicitly correlated CCSD(T)-F12b results have been combined with corrections due to core-valence correlation, scalar-relativistic effects, and higher-order correlation (HC). The latter accounted for effects up to pentuple excitation (CCSDTQP) and was found to be necessary to converge the quality of the PES to the target accuracy of 1 cm${}^{-1}$ in the fundamental vibrational transitions. Comparison with previous results based on multi-reference averaged coupled-pair functional (ACPF) calculations have shown the latter as well as other multi-reference based approaches to be significantly lacking in the accurate description of dynamic HC effects.

Variational calculations employing the present PES and EDMS provide spectroscopic parameters for the vibrational ground state as well as the singly excited states. Comparison with the scarce experimental data shows excellent agreement, i.e., the ground state rotational parameters $B_{0}$ and $D_{0}$ agree to within 0.005% and 0.1%. For the excited states the present spectroscopic parameters should provide reliable estimates for future experimental spectroscopic investigations. In case of the deuterated species DCC${}^{-}$ a 1–3 Darling-Dennison resonance between the CD-stretching state $\left(1,0^{0},0\right)$ and the l=0 component of the *ℓ*-type resonance coupled combination state $\left(0,2^{l},1\right)$ has been identified. However, due to the rather low intensity of the CD as well as the CH fundamental transition $\nu_{1}$ these bands will probably be difficult to detect experimentally. The most promising band $\nu_{2}$ (bending) should be observable around 510 cm${}^{-1}$ and is more intens by a factor of 10–20 compared to the $\nu_{1}$ band. 

## Figures and Tables

**Figure 1 molecules-28-05700-f001:**
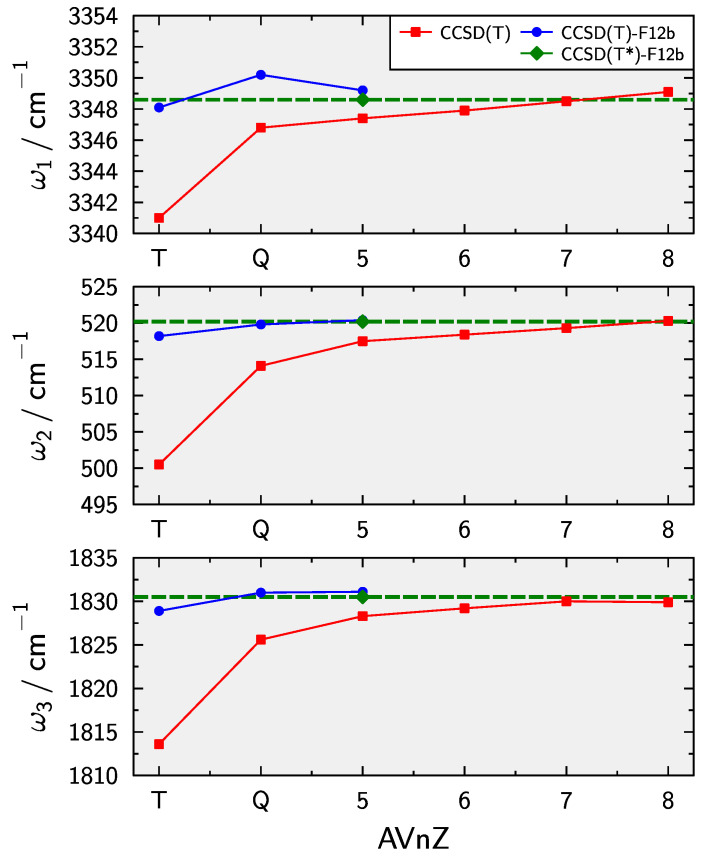
Convergence of conventional CCSD(T) and explicitly correlated CCSD(T)-F12b harmonic vibrational frequencies for HCC${}^{-}$.

**Figure 2 molecules-28-05700-f002:**
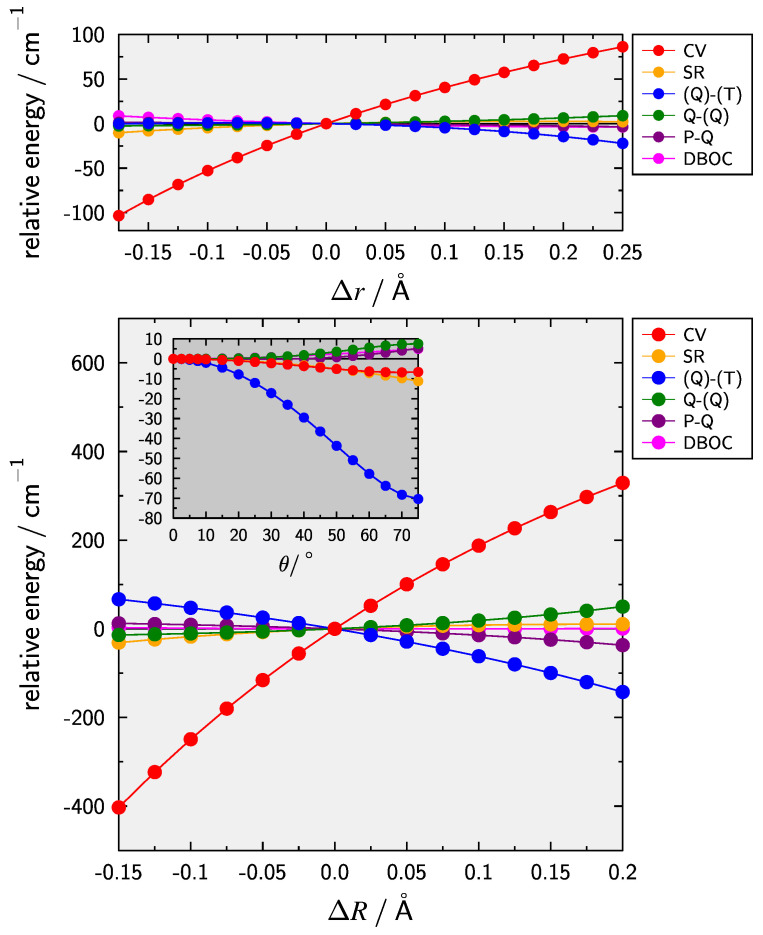
Dependence of smaller contributions to the HCC${}^{-}$ composite PES on the internal coordinates Δr (CH-stretch, upper panel), ΔR (CC-stretch, lower panel), and θ (deviation from linearity, inset lower panel). The DBOC contribution is depicted for the main isotopologue.

**Figure 3 molecules-28-05700-f003:**
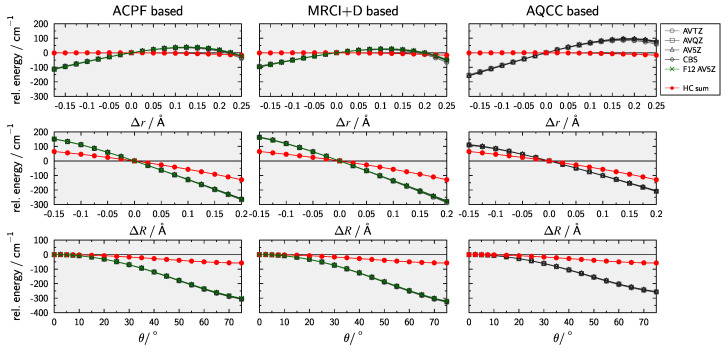
Comparison of dependencies with respect to HCC${}^{-}$ internal coordinates for single reference and multi-reference based methods of calculating higher-order correlation contributions.

**Figure 4 molecules-28-05700-f004:**
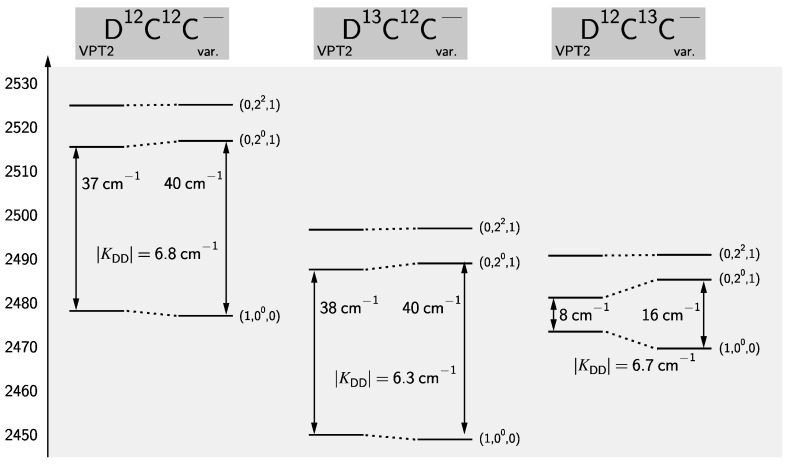
Energy level scheme of $\left(1,0^{0},0\right)$ and $\left(0,2^{l},1\right)$ vibrational states involved in the 1-3 Darling-Dennison resonance in DCC${}^{-}$ isotopologues.

**Figure 5 molecules-28-05700-f005:**
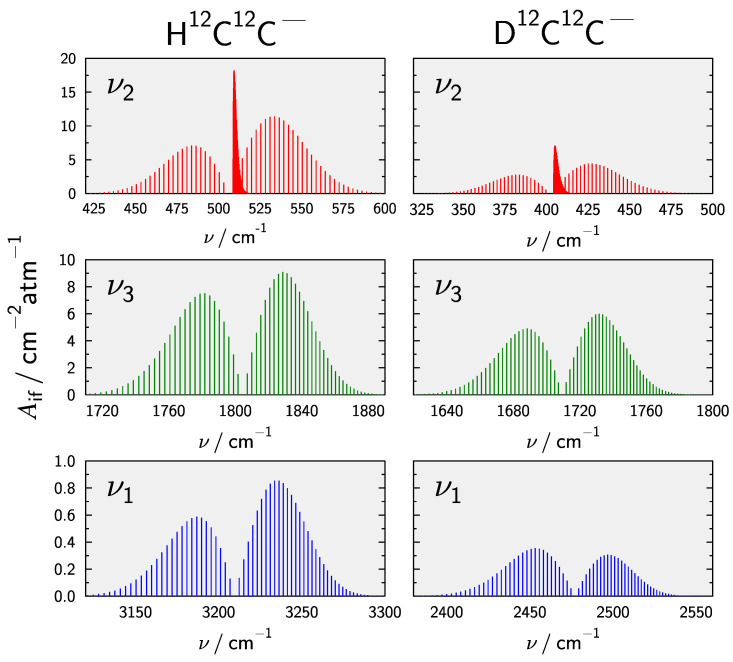
Comparison of stick spectra for the fundamental rovibrational transitions in HCC${}^{-}$ and DCC${}^{-}$at T=300 K. Line intensities are have been calculated using Equation (Equation 9) with g=1 to facilitate a direct comparison.

**Figure 6 molecules-28-05700-f006:**
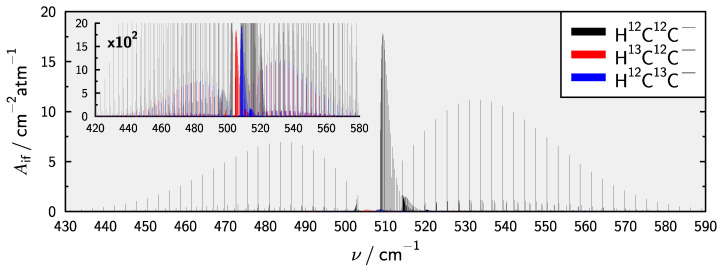
Calculated stick spectrum in the spectral region of the $\nu_{2}$ fundamental for HCC${}^{-}$ and its ${}^{13}$C isotopologues at T=300 K. Additional hot bands originating from the $\left(0,1^{1},0\right)$ state and the $\left(0,2^{l},0\right)$ manifold are also included.

**Table 1 molecules-28-05700-t001:** Frozen-core CCSD(T) and CCSD(T)-F12b equilibrium bond lengths (in Å) and harmonic vibrational frequencies (in cm${}^{-1}$) for HCC${}^{-}$.

Method	Basis	$r_{e}$	$R_{e}$	$\omega_{1}$	$\omega_{2}$	$\omega_{3}$
CCSD(T)	AVTZ	1.07148	1.25467	3341.0	500.5	1813.6
	AVQZ	1.07120	1.25072	3346.8	514.1	1825.6
	AV5Z	1.07087	1.24964	3347.4	517.5	1828.3
	AV6Z	1.07080	1.24931	3347.9	518.4	1829.2
	AV7Z	1.07076	1.24913	3348.5	519.3	1830.0
	AV8Z	1.07071	1.24905	3349.1	520.3	1829.9
CCSD(T)-F12b ^*a*^	AVTZ	1.07082	1.24988	3348.1	518.2	1828.9
	AVQZ	1.07065	1.24909	3350.2	519.8	1831.0
	AV5Z	1.07067	1.24888	3349.2	520.4	1831.1
CCSD(T*)-F12b ^*a*^	AV5Z	1.07072	1.24897	3348.6	520.2	1830.5

^*a*^ Employed geminal exponents are 1.2, 1.4, and 1.5 $a_{0}^{-1}$ for AVTZ, AVQZ, and AV5Z, respectively.

**Table 2 molecules-28-05700-t002:** Dependence of equilibrium bond lengths (in Å) and harmonic vibrational frequencies (in cm${}^{-1}$) on the contributions to the composite PES of HCC${}^{-}$.

Contribution	$r_{e}\;\;$	$R_{e}\;$	$\omega_{1}\;\;$	$\omega_{2}\;\;$	$\omega_{3}\;\;$
F12bs	1.07072	1.24897	3348.6	520.2	1830.5
+CV	−0.00162	−0.00338	+9.2	+4.6	+10.1
+SR	−0.00011	−0.00019	−0.2	−0.2	−0.2
+(Q)–(T)	+0.00011	+0.00084	−3.4	−6.0	−6.6
+Q–(Q)	−0.00008	−0.00022	+1.2	+0.4	+2.4
+P–Q	+0.00004	+0.00018	−0.6	−0.4	−1.8
+DBOC	+0.00011	+0.00001	−0.1	+0.2	+0.1
Composite	1.06917	1.24621	3354.7	518.8	1834.5
Ref. [32]	1.0689	1.2464	3355.4	518.7	1834.8
Ref. [33]	1.06745	1.24702	3356.5	502.3	1828.5
Ref. [34]	1.06945	1.24708	3355.5	524.4	1840.4

**Table 3 molecules-28-05700-t003:** Frozen-core CCSD(T) and CCSD(T)-F12b equilibrium dipole moments (in D) and fundamental band intensities (double harmonic approximation, in km/mol) for HCC${}^{-}$.

Method	Basis	$\mu_{e}$	$A_{01}$	$A_{02}$	$A_{03}$
CCSD(T)	AVTZ	−3.2180	2.27	192.34	71.03
	AVQZ	−3.2163	2.00	187.86	70.62
	AV5Z	−3.2186	1.84	187.28	70.59
	AV6Z	−3.2192	1.79	186.96	70.55
CCSD(T)-F12b ^*a*^	AVTZ	−3.2172	1.87	188.45	70.14
	AVQZ	−3.2188	1.75	186.57	70.11
	AV5Z	−3.2197	1.73	186.89	70.30
CCSD(T*)-F12b ^*a*^	AV5Z	−3.2196	1.77	186.94	70.38

^*a*^ Employed geminal exponents are 1.2, 1.4, and 1.5 $a_{0}^{-1}$ for AVTZ, AVQZ, and AV5Z, respectively.

**Table 4 molecules-28-05700-t004:** Dependence of the equilibrium dipole moment (in D) and fundamental band intensities (double harmonic approximation, in km/mol) on the contributions to the composite EDMF of HCC${}^{-}$.

Contribution	$\mu_{e}\;\;$	$A_{01}\;$	$A_{02}\;\;\;$	$A_{03}\;\;$
F12bs	−3.2196	1.77	186.94	70.38
+CV	−0.0059	−0.25	−1.73	−0.97
+SR	+0.0043	+0.07	+0.63	+0.24
+(Q)–(T)	+0.0017	+0.79	+0.50	+0.06
+Q–(Q)	−0.0001	−0.17	+0.41	+0.12
+P–Q	+0.0010	+0.06	−0.26	−0.25
Composite	−3.2205	2.27	186.49	69.56

**Table 5 molecules-28-05700-t005:** Dependence of spectroscopic parameters obtained from VPT2 on the contributions to the composite PES of HCC${}^{-}$.

Parameter	Contributions							
	F12bs	+CV	+SR	+(Q)–(T)	+Q–(Q)	+P–Q	+DBOC	Comp.
$B_{e}$/MHz	41,632.5	+211.8	+12.2	−49.5	13.5	−10.8	−1.8	41,807.9
$\alpha_{1}$/MHz	294.1	+0.7	+0.4	+1.2	−0.7	+0.2	−0.2	295.7
$\alpha_{2}$/MHz	−130.7	−0.4	−0.1	−0.4	−0.1	+0.2	0.0	−131.5
$\alpha_{3}$/MHz	304.3	+0.5	+0.3	+2.9	−1.6	+1.1	−0.1	307.4
$B_{0}$/MHz	41,464.0	+211.5	+11.9	−51.1	14.9	−11.7	−1.7	41,637.8
$D_{e}$/kHz	93.3	+0.4	+0.1	+0.3	−0.2	+0.1	0.0	94.1
$H_{e}$/MHz	76.5	+1.5	+0.1	−2.3	+1.4	−0.9	0.0	76.3
$q_{2}^{e}$/MHz	246.4	+0.6	+0.2	+1.8	0.0	0.0	−0.1	248.9
$q_{2}^{J}$/kHz	−4.0	0.0	0.0	−0.1	0.0	0.0	0.0	−4.1
$q_{2}^{K}$/kHz	3.8	0.0	0.0	+0.1	0.0	0.0	0.0	3.8
$x_{11}$/cm${}^{-1}$	−59.11	+0.07	−0.04	−0.47	+0.11	−0.03	+0.03	−59.45
$x_{12}$/cm${}^{-1}$	−20.21	+0.02	−0.02	−0.19	+0.03	−0.04	+0.01	−20.40
$x_{13}$/cm${}^{-1}$	−11.78	+0.10	−0.02	−0.20	+0.15	−0.07	+0.01	−11.81
$x_{22}$/cm${}^{-1}$	0.06	−0.11	0.00	+0.08	0.05	+0.04	0.00	0.12
$x_{23}$/cm${}^{-1}$	−5.64	+0.05	0.00	−0.23	+0.03	−0.04	+0.01	−5.84
$x_{33}$/cm${}^{-1}$	−9.09	−0.03	−0.01	−0.12	+0.09	−0.05	0.00	−9.20
$x_{ll}$/cm${}^{-1}$	4.17	+0.05	0.00	−0.04	−0.02	−0.02	0.00	4.15
$\nu_{1}$/cm${}^{-1}$	3204.3	+9.4	−0.3	−4.6	+1.5	−0.7	0.0	3209.5
$\nu_{2}$/cm${}^{-1}$	511.7	+4.3	−0.2	−6.0	+0.6	−0.3	0.2	510.2
$\nu_{3}$/cm${}^{-1}$	1800.8	+10.1	−0.3	−7.2	+2.7	−1.9	0.1	1804.4
$\Delta_{1}$/cm${}^{-1}$	−144.3	+0.2	−0.1	−1.2	+0.3	−0.1	0.1	−145.2
$\Delta_{2}$/cm${}^{-1}$	−8.6	−0.2	0.0	0.0	+0.1	+0.1	0.0	−8.6
$\Delta_{3}$/cm${}^{-1}$	−29.7	0.0	0.0	−0.6	+0.3	−0.2	0.0	−30.2
ZPVE /cm${}^{-1}$	3074.6	+16.5	−0.5	−13.6	+2.4	−1.5	0.2	3078.0
$E_{0}$/cm${}^{-1}$	−2.3	0.0	0.0	0.0	0.0	0.0	0.0	−2.3

**Table 6 molecules-28-05700-t006:** Zero-point vibrational energies (ZPVE, in cm${}^{-1}$) and rotational spectroscopic parameters (in MHz) for the ground vibrational state $\left(0,0^{0},0\right)$ in istopologues of HCC${}^{-}$ obtained from variational calculations. Where available experimental results are provided for comparison ^*a*^.

Isotopologue	Method	ZPVE	$B_{v}$	$10^{3}\;D_{v}$	$10^{6}\;H_{v}$
H${}^{12}$C${}^{12}$C${}^{-}$	this work	3078.0	41,641.6	96.871	0.119
	exp. Ref. [26]		41,639.23501(94)	96.9039(62)	0.13 ^*b*^
H${}^{13}$C${}^{12}$C${}^{-}$	this work	3052.0	40,639.6	92.581	0.106
	exp. Ref. [25]		40,637.441(5)	92.6(2)	
H${}^{12}$C${}^{13}$C${}^{-}$	this work	3058.5	40,113.4	90.080	0.105
	exp. Ref. [25]		40,111.413(7)	90.0(2)	
D${}^{12}$C${}^{12}$C${}^{-}$	this work	2537.0	34,437.6	63.98	0.11
D${}^{13}$C${}^{12}$C${}^{-}$	this work	2508.9	33,893.0	61.88	0.10
D${}^{12}$C${}^{13}$C${}^{-}$	this work	2517.1	33,193.1	59.42	0.100

^*a*^ One standard deviation of the last significant digit is given in parentheses. ^*b*^ Constrained to the theoretical value from Ref. [32].

**Table 7 molecules-28-05700-t007:** Vibrational term energies (in cm${}^{-1}$) and rotational spectroscopic parameters (in MHz) for singly excited vibrational states in istopologues of HCC${}^{-}$ obtained from variational calculations.

	State	$G_{v}$	$B_{v}$	$10^{3}D_{v}$	$10^{6}H_{v}$
H${}^{12}$C${}^{12}$C${}^{-}$	$\left(0,1^{1},0\right)$	510.1	42,773.7	96.871	0.119
	$\left(0,0^{0},1\right)$	1804.5	41,331.9	97.080	0.116
	$\left(1,0^{0},0\right)$	3209.6	41,342.6	96.183	0.125
H${}^{13}$C${}^{12}$C${}^{-}$	$\left(0,1^{1},0\right)$	506.0	40,756.9	95.167	0.136
	$\left(0,0^{0},1\right)$	1776.3	40,340.1	92.777	0.103
	$\left(1,0^{0},0\right)$	3195.4	40,361.3	91.932	0.110
H${}^{12}$C${}^{13}$C${}^{-}$	$\left(0,1^{1},0\right)$	509.3	40,238.1	92.743	0.136
	$\left(0,0^{0},1\right)$	1768.8	39,820.0	90.268	0.102
	$\left(1,0^{0},0\right)$	3208.7	39,828.8	89.445	0.108
D${}^{12}$C${}^{12}$C${}^{-}$	$\left(0,1^{1},0\right)$	405.5	34,607.1	67.402	0.173
	$\left(0,0^{0},1\right)$	1709.5	34,225.5	64.006	0.110
	$\left(1,0^{0},0\right)$	2477.0	34,153.0	66.860	0.470
D${}^{13}$C${}^{12}$C${}^{-}$	$\left(0,1^{1},0\right)$	400.2	34,049.6	64.973	0.154
	$\left(0,0^{0},1\right)$	1693.3	33,684.3	61.909	0.095
	$\left(1,0^{0},0\right)$	2448.9	33,624.1	64.164	0.390
D${}^{12}$C${}^{13}$C${}^{-}$	$\left(0,1^{1},0\right)$	405.5	34,607.1	67.402	0.173
	$\left(0,0^{0},1\right)$	1677.4	32,991.2	59.454	0.097
	$\left(1,0^{0},0\right)$	2469.7	32,986.8	86.141	3.604
	**State**		$q_{v}$	$10^{3}q_{v}^{J}$	$10^{6}q_{v}^{\mathrm{JJ}}$
H${}^{12}$C${}^{12}$C${}^{-}$	$\left(0,1^{1},0\right)$		258.3	−4.727	0.101
H${}^{13}$C${}^{12}$C${}^{-}$	$\left(0,1^{1},0\right)$		248.0	−4.302	0.087
H${}^{12}$C${}^{13}$C${}^{-}$	$\left(0,1^{1},0\right)$		240.2	−4.196	0.086
D${}^{12}$C${}^{12}$C${}^{-}$	$\left(0,1^{1},0\right)$		220.5	−4.431	0.114
D${}^{13}$C${}^{12}$C${}^{-}$	$\left(0,1^{1},0\right)$		216.3	−4.170	0.104
D${}^{12}$C${}^{13}$C${}^{-}$	$\left(0,1^{1},0\right)$		220.5	−4.431	0.114

## Data Availability

The data that support the findings of this study are available from GRO.data (doi:10.25625/YBLAAU) [132]. This includes the raw ab initio points used in the construction of the composite potential energy surfaces and electric dipole moment surface as well as the outputs of the C8vpro program [109] used for the variational calculations. Within the latter the lowest 90 eigenvalues of the rovibrational Hamiltonian up to $J_{\max}=60$ in both *e* and *f*-parity can be found. Furthermore, a line list for a number of rotational and rovibrational bands in the main isotopologue of HCC${}^{-}$ has also been deposited with GRO.data. Additional results are included in the article Appendix A (for details see above).

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
