# Peer review of "Ab Initio Rovibrational Spectroscopy of the Acetylide Anion"

_molecules, 2023, doi:10.3390/molecules28155700_

Round 1

Reviewer 1 Report

The manuscript "Ab initio rovibrational spectroscopy of the acetylide anion" describes the construction of potential energy surface of HCC- anion. PES is constructed using coupled-clusters theory and several contribution into the calculated values. The influence of hydrogen-carbon and carbon-carbon distances as well as the angle HCC on the total energy is thoroughly studied. The simulated rovibrational spectra show excellent agreement with the previously reported experimental and the most reliable calculated data. Author also provides a very informative literature review on the topic and gives the practical recommendations how the acetylide ion should be best detected in the interstellar medium, which would come handy for the astronomers. The precision of author's method is immense. The manuscript is nearly perfect, and there only some typos that add the word "nearly" ("bechmark", line 85, "correlcation", line 107, "intens", line 510, etc.)

Author Response

Reviewer Comment:
The manuscript is nearly perfect, and there only some typos that add the
word "nearly" ("bechmark", line 85, "correlcation", line 107, "intens",
line 510, etc.)
Response:
The three typos mentioned by the reviewer have been corrected.

Reviewer 2 Report

This contribution details a theoretical investigation into the acetylide anion, first building a state-of-the-art potential energy surface, then performing rovibrational calculations to predict spectra. The chosen system is relevant to astrochemistry and considering anions always adds an additional degree of difficulty to electronic structure calculations. The composite method chosen to produce the surface(s) is of very high quality and it is difficult to see how it could be improved upon with current methods. The rovibrational calculations are also carried out using appropriate methods, and comparison to what experimental data there are available is excellent. The manuscript is well-written and easy to follow. I have a minor suggestion to improve the referencing of basis sets, and spotted a few very minor typographical errors, which the author may like to address before publication.

Referencing.

On lines 110 and 137, the author cites reference 65 for core-valence correlating basis sets, but this manuscript is one describing diffuse augmented sets. I would recommend adding a reference to https://doi.org/10.1063/1.470645 

I would also suggest changing "Dunning-type" on line 116 to "correlation consistent-type", but this is something of a personal preference.

Typos:

Line 148: "lenghts" should be "lengths".

Line 281: "beyound" should be "beyond".

Line 288: "differenc" should be "difference".

Line 331: "an" should be "and".

Line 336: "accurated" should be "accurate".

Author Response

Reviewer Comment:
On lines 110 and 137, the author cites reference 65 for core-valence
correlating basis sets, but this manuscript is one describing diffuse
augmented sets. I would recommend adding a reference to https://doi.org/10.1063/1.470645

Response:
Thank you for pointing this obvious error out. The reference has
been changed to the one suggested.

Reviewer Comment:
I would also suggest changing "Dunning-type" on line 116 to "correlation
consistent-type", but this is something of a personal preference.

Response:
The wording "Dunning-type" has been removed entirely.

Reviewer Comment:
• Line 148: "lenghts" should be "lengths".
• Line 281: "beyound" should be "beyond".
• Line 288: "differenc" should be "difference".
• Line 331: "an" should be "and".
• Line 336: "accurated" should be "accurate".

Response:
All the above mentioned typos have been corrected.
